# Differentially Expressed microRNAs in MIA PaCa-2 and PANC-1 Pancreas Ductal Adenocarcinoma Cell Lines are Involved in Cancer Stem Cell Regulation

**DOI:** 10.3390/ijms20184473

**Published:** 2019-09-10

**Authors:** Ye Shen, Kefeng Pu, Kexiao Zheng, Xiaochuan Ma, Jingyi Qin, Li Jiang, Jiong Li

**Affiliations:** 1Key Laboratory for Nano-Bio Interface Research, Nano-Bio-Chem Centre, Suzhou Institute of Nano-Tech and Nano-Bionics, Chinese Academy of Sciences, Suzhou 215123, China; 2State Key Laboratory of Radiation Medicine and Protection, School for Radiological and interdisciplinary Sciences (RAD-X), Collaborative Innovation Center of Radiation Medicine of Jiangsu Higher Education Institutions, Soochow University, Suzhou 215123, China

**Keywords:** pancreatic ductal adenocarcinoma (PDAC), microRNA, cancer stem cell (CSC), CSC surface marker, tumor sphere formation

## Abstract

Pancreatic ductal adenocarcinoma (PDAC) is one of the most lethal malignancies, and thus better understanding of its molecular pathology is crucial for us to devise more effective treatment of this deadly disease. As cancer cell line remains a convenient starting point for discovery and proof-of-concept studies, here we report the miRNA expression characteristics of two cell lines, MIA PaCa-2 and PANC-1, and discovered three miRNAs (miR-7-5p, let-7d, and miR-135b-5p) that are involved in cancer stem cells (CSCs) suppression. After transfection of each miRNA’s mimic into PANC-1 cells which exhibits higher stemness feature than MIA-PaCa-2 cells, partial reduction of CSC surface markers and inhibition of tumor sphere formation were observed. These results enlighten us to consider miRNAs as potential therapeutic agents for pancreatic cancer patients via specific and effective inhibition of CSCs.

## 1. Introduction

Pancreatic ductal adenocarcinoma (PDAC), which constitutes 90% of all pancreatic cancers, is one of the most deadly malignant diseases, with an overall five-year survival rate of 8% for all stages combined and is ranked as the fourth leading cause of cancer-related death in the world [1,2]. The dismal prognosis is mainly due to late diagnosis and therapeutic resistance, which contributes to tumor local recurrence and distant metastasis [2,3]. Furthermore, individual differences of patients increase the complexity of PDAC treatments. As a consequence, there is an urgent need to better understand the molecular pathology of PDAC.

Four well-known cancer genes (*KRAS*, *TP53*, *SMAD4* and *CDKN2A*) act as oncogenes in PDAC [4]. Yet, none has been proved effective anti-cancer drug target after decades of research. An alternative is to seek anticancer molecules acting on other components of the genome, such as epigenetic factors [5]. Currently, microRNAs (miRNAs) are attracting remarkable research attention because of their association with tumor growth, invasion, angiogenesis, and immune evasion. miRNAs are small (19–24 nucleotides) noncoding RNA molecules that primarily function to repress target mRNA translation. At least 50% of all miRNAs are aberrantly expressed in tumors and exhibit oncogenic or tumor suppressive activities [6].

Many studies have shown that miRNAs play a critical role in the regulation of cancer stem cells (CSCs) in malignant tumors, including PDAC [7,8]. CSCs possess the ability to self-renew and persist in tumors as a specific cancer cell subpopulation that can contribute to relapse and metastasis by reemerging or initiating new tumors [9,10,11]. Since CSCs are closely associated to therapeutic resistance and disease recurrence, visionary therapeutic strategies are proposed to develop clinically useful inhibitors to target the CSCs [12]. Hence, it is of great value to explore the possibility of CSC-suppressive miRNAs as novel small molecule tools for tumor therapy.

MIA PaCa-2 and PANC-1 are two commonly used tumor cell lines for in vitro studies of PDAC carcinogenesis [13]. They are easily accessible, reliable, and less problematic compared to the primary culture of tumors. According to Deer et al. [14], both MIA PaCa-2 and PANC-1 are poorly differentiated, and PANC-1 is derived from a patient with metastasis while the metastatic state of the patient from whom MIA PaCa-2 is derived is unclear. The invasive assays of these two cell lines conducted by different research groups do not produce a consensus and solid result. However, in tumorigenicity assay, MIA PaCa-2 tumors are larger than PANC-1 when Severe Combined Immunodeficiency (SCID) mice receive intraperitoneal injections of equal dose of tumor cells. Moreover, their morphological and genetic characteristics are also well studied [13], suggesting these cell lines are ideal tools for pancreatic cancer research in vitro.

The miRNA expression profiling data of PDAC cell lines from a previous study revealed some dysregulated miRNAs that enhance proliferation or growth arrest, however only 321 human miRNAs are monitored in the custom microarray [15], and the authors focused on the uniform pattern of up- or down- regulated miRNA between PDAC cell lines and control non-transformed pancreatic ductal cell lines. In this work, we attempt to seek the heterogeneity between different cancer cell lines, as “precision therapeutics” has specific concern on the individual characteristics. Using our fluorescent label-free microarray [16] contained 2023 human miRNA sequences, we identified differentially expressed miRNAs between MIA PaCa-2 and PANC-1 cell lines. Interestingly, bioinformatic analysis revealed that the target genes of three down-regulated miRNAs (miR-7-5p, miR-135b-5p and let-7d) in PANC-1 cell line are related to stem cells signaling pathway. Currently, identification of cancer stem cells is mainly dependent on surface markers that are shared by normal stem cells, meanwhile sphere-forming assays that have been widely used to identify stem cells to evaluate self-renewal and differentiation at the single cell level in vitro [12,17]. Using these techniques, we observed that overexpression of the three identified miRNAs in PANC-1 cell line resulted in the partial CSC surface markers down-regulation and inhibition of tumor sphere formation.

At present, some miRNAs and anti-miRNA constructs are rising as potential drug treatments for cancers [18]. For example, one antimiR-122 trial for hepatitis C therapy has reached in phase IIa [19]. Our preliminary results here provide novel information related to potential miRNA therapeutic agents for the inhibition of CSCs, and may lead to a strategy to treat PDAC patients with more stem-like cancer cells as an auxilliary approach to enhance main therapies.

## 2. Results

### 2.1. Screening of Differentially Expressed miRNAs between MIA PaCa-2 and PANC-1 Cell Lines

The differentially expressed miRNAs between MIA PaCa-2 and PANC-1 cell lines were screened using a human miRNA microarray. A total of 33 differentially expressed miRNAs were identified, including 28 up-regulated and 5 down-regulated miRNAs in the PANC-1 cell line compared to MIA PaCa-2 (Appendix A). Hierarchical clustering analysis revealed a clearly distinct expression of these 33 differentially expressed miRNAs between MIA PaCa-2 and PANC-1 cell lines (Figure 1A).

### 2.2. Validation of Differentially Expressed miRNAs by Real-Time RT-PCR

According to the data of miRNA array, 33 differentially expressed miRNAs were selected for validation by real-time RT-PCR using specific primers (Appendix A). We confirmed the up-regulation of 13 miRNAs (miR-100-5p, miR-31-5p, miR-99a-5p, miR-376b-3p, miR-495-3p, miR-196a-5p, miR-9-5p, miR-487b-3p, miR-138-5p, miR-127-3p, miR-154-3p, miR-424-5p, miR-181b-5p) and down-regulation of 5 miRNAs (miR-221-3p, let-7d-5p, miR-7-5p, miR-4468, miR-135b-5p). The dynamic expression patterns obtained with real-time RT-PCR were consistent with the microarray results (Figure 1B).

### 2.3. Target Prediction, Gene Ontology (GO) and Pathway Enrichment Analysis for miRNAs Significantly Changed in PDAC Cell Lines

Putative target genes of validated miRNAs that differentially expressed in MIA PaCa-2 and PANC-1 cell lines were shown in Appendix A. The functional enrichment analysis revealed the enriched GO terms in the predicted target genes of miRNA were associated with molecular functions (MF) like protein binding and transcription factor binding, biological characteristics (BP) including positive and negative regulation of transcription from RNA polymerase II promoter, and the gene products (CC) were primarily found in nucleus and nucleoplasm (Figure 2A, Appendix A). The enriched pathways were revealed by Kyoto Encyclopedia of Genes and Genomes (KEGG) analysis. Among the top ten pathways mostly likely to be regulated by the differentially expressed miRNAs between these two PDAC cell lines are signaling pathways that regulating pluripotency of stem cells, Forkhead box O (FoxO) signaling pathway, ErbB signaling pathway and Hypoxia-inducible factor 1 (HIF-1) signaling pathway, etc. (Figure 2B, Appendix A).

### 2.4. Interaction Network Analysis

To identify the putative functional modules, we constructed the miRNA gene network (Figure 3) based on the targets genes simultaneously involved in enriched functions and pathways and the miRNA–target gene binding information. The network pointed out several miRNAs with a high degree of enrichment, including miR-9-5p, miR-7-5p, let-7d, miR-138-5p and miR-424-5p. They had higher degree of regulation in the miRNA gene network, with 30, 25, 17, 15 and 14 target genes, respectively. By contrast, miR-154-3p, miR-487b-3p, miR-99a-5p, miR-376b-3p and miR-181b-5p, had only 1 or 2 target genes.

Interestingly, we found the target genes of miR-7-5p (such as *RAF1*, *PIK3CD*, *PAX6*, *PIK3R3*, *PIK3CB*, *IGF1R*, *BMPR2*, *AKT3*), miR-135b-5p (such as *SMAD5*, *KLF4*, *APC*, *ACVR1B*), and let-7d (such as *PCGF3*, *HAND1*, *DVL3*) were enriched in the signaling pathways regulating the pluripotency of stem cells (Appendix A). As these three miRNAs were down-regulated in PANC-1 cell lines, the expression level of these stem cell related genes were probably elevated in PANC-1 cell lines since miRNAs were known as negative regulator of their target genes.

### 2.5. Differences in the Stem Cell Surface Markers Expression Percentage of PANC-1 and MIA-PaCa-2

Noting that the potential target genes enriched in the signaling pathways are involved in stem cell pluripotency regulation and are probably upregulated in PANC-1 cell lines, we analyzed the expression patterns of pancreatic CSC surface markers CD24, CD44, and EpCAM (17) in PANC-1 and MIA-PaCa-2. The flow cytometry analysis results illustrated that in both cell lines, all cells showed CD44 expression. In PANC-1 cell line, the percentage of CD24^+^ cells was 9.8% and EpCAM^+^ cells was 36.6%. In addition, the percentage of CD24^+^CD44^+^EpCAM^+^ cells in PANC-1 was 7.2%. While in MIA-PaCa-2 cell line, CD24^+^ cells and EpCAM^+^ cells were not detected (Figure 4A).

To confirm the markers expression results, the presence of CD24, CD44, and EpCAM on the undetached cells was also determined by immunocytochemistry. The results of PANC-1 and MIA-PaCa-2 were similar to the trend of flow cytometry analysis that more stem cell surface markers expressed in PANC-1. As different method detection, there are some difference on details. In PANC-1 and MIA-PaCa-2 cell lines, 99.9% cells were CD44 positive. In PANC-1, the percentage of CD24^+^ cells was 17.3 ± 3.7% and EpCAM^+^ cells was 46.0 ± 3.5%. In MIA-PaCa-2, the percentage of CD24^+^ cells was 11.5 ± 0.7% and EpCAM^+^ cells was 38.7 ± 0.5%. The percentage of CD24^+^CD44^+^EpCAM^+^ cells in PANC-1 was 16.8 ± 3.6% and in MIA-PaCa-2 was 11.5 ± 0.7% (Figure 4B,C). These results of stem cell surface markers detection indicated PANC-1 cell line is more “stemness” than MIA-PaCa-2.

### 2.6. Differences in Capacity of Sphere Formation between PANC-1 and MIA-PaCa-2

Development of tumor spheres is often associated with CSC phenotype [4]. PANC-1 and MIA-PaCa-2 cells were cultured in SFM to assess the differences in the ability of sphere formation (Figure 5). The results illustrated that the percentage of sphere formation rate in PANC-1 was 96.7%, in detail 28.3 ± 0.6 spheres formed in the 30 seeded wells. Compared to PANC-1, the percentage of sphere formation rate in MIA-PaCa-2 was 80%, about 25.3 ± 1.2 spheres formed in the 30 seeded wells. It also proved PANC-1 cell line owned more cancer stemness.

### 2.7. The Effect of miRNA Mimics on the Stem Cell Surface Markers Expression Percentage of PANC-1

As miR-7-5p, miR-135b-5p, and let-7d predicted involving in stem cell related signaling pathway and validated down-regulate in PANC-1 cell line, we transfected these three synthetic miRNA mimic constructs using lipofectamine 2000 to delivere into the PANC-1 and evaluate their CSC characteristics. The results of the CSC surface marker expression by flow cytometry analysis revealed the expression percentage of CD44 was decreased from 99.6% to 98.1% (miR-7-5p), 95.9% (let-7d), 98.6% (miR-135b-5p) and 94.2% (three mimics mixture) after PANC-1 transfected with miRNA mimics (Figure 6A). Meanwhile, the percentage of EpCAM in PANC-1 was changed from 32.4% to 32.7% (miR-7-5p), 27.8% (let-7d), 26.2% (miR-135b-5p) and 27.4% (three mimics mixture), and the percentage of CD24 down-regulated from 10.7% to 5.9% (miR-7-5p), 6.2% (let-7d), 6.6% (miR-135b-5p) and 6.1% (three mimics mixture). In addition, the percentage of CD24^+^CD44^+^EpCAM^+^ cells was 7.4% in PANC-1, but was 4.3% (miR-7-5p), 4.0% (let-7d), 4.1% (miR-135b-5p) and 3.9% (three mimics mixture) in PANC-1 transfected with miRNA mimics (Figure 6B).

Furthermore, the presence of CSC-related surface markers was also determined by immunocytochemistry (Figure 7A). Different from the results of flow cytometry analysis that the expression of surface marker altered slightly, after PANC-1 treated with miRNA mimics, the percentage of CD24^+^ cells, EpCAM^+^ cells and CD24^+^CD44^+^EpCAM^+^ cells almost halved by immunocytochemistry tests (Figure 7B,D,E). Especially for PANC-1 cell line transfected with let-7d mimic, the CD24^+^, EpCAM^+^ and CD24^+^CD44^+^EpCAM^+^ cells decreased from 22.6 ± 2.5%, 52.3 ± 1.9% and 22.4 ± 2.4% to 6.0 ± 2.9%, 21.0 ± 6.2% and 5.9 ± 2.9%. However, the percentage of CD44^+^ cells did not evident change (Figure 7C). These results of surface marker analysis indicated upregulation of miRNAs (miR-7-5p, let-7d and miR-135b-5p) in PNAC-1 could inhibit the CSC-related surface marker expression.

### 2.8. The Regulation of miRNA Mimics on the Sphere Formation Ability in PANC-1

Following the expression of stem cell surface markers analyzed, we checked the sphere formation ability of PANC-1 with miRNA mimics (miR-7-5p mimic, miR-135b-5p mimic, let-7d mimic and negative control mimic) transfected. Results showed that all groups of PANC-1 with different treatments could form spheres. However, the blank and control groups formed tighter 3D spheres with higher sphere roundness (Figure 8). On the day 1 and 3, the sphere roundness showed no significant difference. While on the day 5, the miR-7-5p mimic, let-7d mimic, miR-135b-5p mimic and three mimics mixture treated groups showed looser 3D spheres with low sphere roundness. The sphere roundness was 0.28 ± 0.07, 0.29 ± 0.1, 0.25 ± 0.12 and 0.33 ± 0.10 for miR-7-5p, let-7d, miR-135b-5p and three mixture mimics treated, respectively, while blank and control group was 0.49 ± 0.06 and 0.46 ± 0.06 (Figure 8B). The results illustrated that the sphere formation ability (sphere roundness) declined in PNAC-1 when miRNA mimics treated, but no significant difference between single mimic and three mimics mixture transfected group.

## 3. Discussion

In this study, we have confirmed 18 (13 up- and 5 down-regulated) differentially expressed miRNAs between MIA PaCa-2 and PANC-1 cell lines, and have discovered these miRNAs are also involved in the development of several tumors. For example, miR-7-5p has been found down-regulated in glioblastoma multiforme (GBM) [20], while up-regulated in oral cancer [21]. Meanwhile, miR-99a-5p and miR-196a-5p are strongly associated with the presence of the tumor via the analysis of the TCGA HNSC dataset [21]. In comparison of cancerous and noncancerous tissues of PDACs, miR-31-5p and miR-100-5p show up-regulation and their functions are to promote proliferation, invasion and migration, or inhibit proliferation and increase sensitivities to cisplatin [6,22]. In addition, miR-181b-5p is up-regulated in colorectal cancer and breast cancer by targeting programmed cell death 4 (PDCD4) and Bim, respectively [23,24]. Furthermore, its expression is also probably correlated with burkitt’s lymphoma (BL) [25].

Previous studies compared miRNA expression between CSC surface marker positive cells and parental cells in cancer cell lines [26,27,28] in attempt to discover novel miRNA markers capable of discerning CSCs, which may also serve as potential therapeutic targets of CSCs. For instance, Bin et al. have studied the aberrant expressed miRNA of CSC surface marker positive cells from human pancreatic cancer cell lines, and find that knockdown of miR-125b results in the inhibition of tumor cell aggressiveness of CSCs [8]. In this study, we focused on the different expressed miRNAs between two PDAC cell lines, PANC-1 and MIA-PaCa-2, to illustrate the molecular pathology of PDAC, and better understand these cell lines, which probably reflect the individual differences of patients.

Here, via bioinformatics analysis, some predicted targets of the differentially expressed miRNAs between PANC-1 and MIA-PaCa-2 are enriched in signaling pathways which regulate the pluripotency of stem cells. It is well known that CSCs share gene networks with pluripotent and adult stem cells, as well as these gene networks are essential for self-renewal and pluripotency. Thus, we analyzed CSC surface markers and the capacity of tumor sphere formation to validate the prediction. More CSC surface markers CD24 and EpCAM were detected in PANC-1, although 99.9% cells were CD44 positive in both cell lines. Similar results were seen from Gradiz et al. that MIA-PaCa-2 was CD24^−^ and CD44^+^, and PANC-1 was CD24^−/+^ and CD44^+^, while in their work EpCAM was not interrogated [13]. On the other hand, we found that PANC-1 exhibits stronger sphere formation than MIA-PaCa-2. These results indicate that the PANC-1 cell line possesses higher “stemness” feature.

The existence of CSCs has been widely accepted to be responsible for tumor aggressiveness. In PDAC, CSCs may display additional properties, such as relative drug resistance and enhanced invasion and migration. Emerging evidences suggest that miRNAs may play a key role in the biology of CSCs. A series of miRNAs has been found aberrantly expressed in PDAC CSCs, including miR-99a-5p, miR-100-5p, miR-125b, miR-192, and miR-429 [29]. In the presented study, we have also detected miR-99a-5p and miR-100-5p with differential expression between MIA-PaCa 2 and PANC-1 cell lines. Many previous studies have demonstrated that various miRNAs participate in promoting or inhibiting stemness. For instance, the miR-200 family members repress the expression of stem cell factors Y–box 2 (Sox2) and Kruppel-like factor 4 (Klf4). The epithelial-to-mesenchymal transition (EMT)-activator zinc finger E-box binding homeobox 1 (ZEB1) represses the expression of stemness-inhibiting miR-203. These results show that the ZEB1-miR-200 axis is probably a novel therapeutic target against PDAC CSCs [11,30]. Several other studies also have identified some miRNAs with important roles in cancer stemness in PDAC, such as miR-106a, miR-30a, miR-744 and miR-1246 [31,32,33,34]. Moreover, let-7c is shown to decrease pancreatic cancer initiating cell growth by posttranscriptional activation of Numbl and indirect inhibition of Notch [35], which is similar to our results that let-7d mimic partially reduces CSC surface markers and inhibits tumor sphere formation. In other cancers, Sanchez-Diaz et al. have verified the ability of tumor sphere formation is impaired when the expression of hsa-miR-135b-5p is suppressed in SK-N-BE (2) (neuroblastoma) cells [36], the same as we observed in PDAC PANC-1 cells.

Other signaling pathways potentially targeted by these differentially-expressed miRNAs between PANC-1 and MIA-PaCa-2 are also of strong correlation to cancers. One among them is the FoxO signaling pathway, which has critical roles in a number of physiological and pathological conditions including cancer [37]. FoxO transcription factors have been considered as tumor suppressors due to their pro-apoptotic and anti-proliferative functions [38]. Kumazoe et al. have confirmed that FoxO3 depletion is sufficient to suppress the CSC phenotype of PDAC cells [39]. Meanwhile, the ErbB signaling pathway is associated with the development of a wide variety of solid tumors. ErbB-1 (also known as EGFR, epidermal growth factor receptor) and ErbB-2 (also known as HER2, human epidermal growth factor receptor 2), are famous therapeutic targets for cancer treatment and their excessive signaling play critical roles in the development and malignancy of many tumors. For example, in preclinical models of pancreatic cancer, targeting of ErbB results in enhanced antitumor activity [40]. In addition, HIF-1 signaling pathway plays an integral role in the body’s response to low oxygen concentrations, and involves in tumor proliferation due to its role in hypoxia [41]. Taken together, it occurs reasonable to assume that the different expression of miRNAs are accountable for the phenotypic discrepancies of the two PDAC cell lines.

The mimics of miR-7-5p, miR-135b-5p and let-7d were used to transfect into PANC-1 cells, respectively, and they all partially suppressed the sphere formation. Previous researches have indicated that miR-7-5p, miR-135b-5p and let-7d are characterized as both oncogenes and tumor suppressors, depending on the specific context of various tissues. miR-7-5p is more of a tumor suppressor that represses oncogenic proteins such as EGFR, IGF1R, IRS-1, IRS-2, RAF1, PIK3CD, mTOR, and PI3K, in various cancers including PDAC, liver, breast, non-small cell lung carcinoma (NSCLC), colorectal (CRC) and ovarian [42]. Recent research have revealed that miR-7-5p targets SOX18 to inhibit tumor cell proliferation, migration and invasion in PDAC [43]. Similarly, let-7d mainly acts as a tumor suppressor, and has a significant impact on EMT and formation of cancer initiating cells [44]. Conversely, miR-135b-5p promotes migration, invasion and EMT of pancreatic cancer cell by targeting NR3C2 as an oncogene, and overexpression of miR-135b-5p leads to poor prognosis [45]. We also discovered PANC-1 cells co-transfected with 3 mimics exhibited similar sphere-forming efficiency with single mimic transfected. It is probably because that the three miRNAs target many molecules that involve in different pathways, and the overall outcome may be the balance among them. Therefore, more work about the targets of these miRNAs needed to be done in the further studies.

Previously, microscopic observation of PANC-1 and MIA PaCa-2 cells have revealed there are two distinct morphological patterns (large cells and small cells) in MIA PaCa-2, while three distinct morphological patterns (large cells, intermediate cells and small cells) in PANC-1 [13]. Furthermore, it is confirmed via immunohistochemistry analysis that E-cadherin expressed in MIA PaCa-2 but not in PANC-1. Since increased expression of E-cadherin is associated with improved survival in several tumor types [46], the behavior of PANC-1 is considered more aggressive with a greater metastasizing potential. These results indicate that PANC-1 is more heterogenous and invasive than MIA PaCa-2, which is in agreement to our results that PANC-1 cells own more cancer stemness.

miRNAs have been viewed as possible therapeutic targets since these molecules affect multiple gene expression and cellular pathways. For instance, transfection of miR-200c and miR-200a mimics into pancreatic CSCs could reduce their colony formation, invasiveness, and resistance to chemotherapy by regulating EMT [47]. When miRNA-205 and miR-7 were transfected into pancreatic cells resistant to gemcitabine, the expression of tubulin beta 3 class III (TUBB3) and p21 activated kinase 1 (Pak-1) were reduced, respectively, and the CSC population was also decreased [48]. These researches support our view that targeting miRNAs could become a novel strategy for killing CSCs, which may become a powerful assistance to the treatment of patients diagnosed with PDAC.

To find miRNA dysregulation in cancers through comparing cancerous vs. normal cells, tissues, or plasma is a conventional way to analyze the tumor-related signal pathways. Nonetheless, inter-cell/patient variability of miRNA expression and signaling pathway is rarely investigated. In this work, we have demonstrated that in pancreatic ductal adenocarcinoma cell lines, PANC-1 cells exhibit higher stemness feature than MIA-PaCa-2 cells, which is probably the consequence of differential regulation of specific miRNAs. Confirmatory assays using the corresponding miRNA mimics show inhibition of the CSC-related characteristics in PANC-1. These findings may offer new therapeutic targets for precise and personalized treatment of patients, or offer an alternative approach to enhance the efficacy of current therapies. For instance, to suppress tumor growth by restoring miRNA expression to kill CSCs. However, the study of the above miRNA-based therapies is still in its infancy, more researches are required to verify the effectiveness of specific therapeutic targets for individual differences.

## 4. Materials and Methods

### 4.1. Cell Culture

The human pancreatic adenocarcinoma cell lines PANC-1 and MIA-Paca-2 were obtained from the Cell Bank of Type Culture Collection of Chinese Academy of Sciences. These two cells were cultured in Dulbecco’s modified Eagle’s high glucose medium (DMEM, Gibco, Gaithersburg, MD, USA) supplied with 10% (*v/v*) fetal bovine serum (FBS, HyClone, Logan, UT, USA) at 37 °C in a 5% CO_2_ humidified environment.

### 4.2. Extraction of microRNA and Microarray Screening

Total RNA including miRNAs from MIA Paca-2 and PANC-1 cell lines were extracted using miRNeasy Micro Kit (Qiagen, Hilden, Germany) according to the manufacturer’s instructions. The miRNA microarray platform was used as previously described using a fluorescent label-free strategy with good data reproducibility, highly specificity and high sensitivity [16]. These microarrays were custom-made by Phalanx Biotech Group (Hsinchu, Taiwan, China) and each probe printed in quadruplicate. The method of chip activation was followed the company’s protocol, and the procedure of chip hybridization was described as follows. Total RNA (4.0 ug) from each sample were hybridized onto the miRNA microarray, which included 2023 mature human miRNA sequences (Sanger miRBase, release 19.0). Briefly, total RNA from each sample and 200 nM (final concentration) Cy3-labeled reporter molecule (Universal Tag, UT) were dissolved in hybridization buffer, and 4 non-human miRNAs were spiked-in as an external control. The array was hybridized at 44 °C for 48 h in a hybridization oven (2545A, Agilent, Santa Clara, CA, USA) with a constant rotation speed of 15 rpm. After hybridization, slides were washed and scanned using a LuxScan 10 K Microarray Scanner (CapitalBio, Beijing, China) at constant power and PMT gain settings through a single-color channel (532 nm wavelength). Finally, raw data were collected with GenePix Pro 7.0 software package (Axon, Union City, CA, USA). Differently expressing miRNAs were selected using the paired *t*-test with the cut-off criteria of *P* < 0.01 and |fold change| > 3 for further analysis.

### 4.3. miRNA Expression Analysis Using qRT-PCR

The microarray data was validated by qRT-PCR. Extracted total RNA (500 ng) from each sample was reverse transcribed into cDNA using the PrimeScript™ RT reagent Kit (RR037A, TaKaRa, Shiga, Japan) with specific stem-loop primer for miRNA. Human small nuclear U6 RNA was used as the internal reference for normalization. Real-time qPCR was performed to evaluate the expression levels of mature miRNAs on a Roche LightCycler System (Roche Diagnostics, Rotkreuz, Switzerland) using SYBR Premix Ex Taq™ II (RR820A, TaKaRa, Shiga, Japan). Cycling conditions were as follows: 95 °C for 30 s, 95 °C for 5 s and 60 °C for 50 s, followed by 40 cycles. Melting curves were generated for each qRT-PCR reaction to verify the specificity. All the reactions were performed in triplicate and relative fold changes were calculated by the equation 2^−∆∆*C*t^. The sequences of the primers used in qRT-PCR were listed in Appendix A.

### 4.4. Bioinformatics

The miRNA targets were predicted by at least two databases of the following prediction databases: TargetScan (Whitehead Institute for Biomedical Research, Cambridge, MA, USA, http://www.targetscan.org/vert_72/), miRanda (Memorial Sloan-Kettering Cancer Center, New York, NY, USA, http://www.microrna.org/microrna/getDownloads.do) and miRTarBase (National Chiao Tung University, Hsinchu, Taiwan, http://mirtarbase.mbc.nctu.edu.tw/php/index.php). The Gene Ontology (GO) functional and pathway enrichment analysis were conducted for the target genes using the Database for Annotation, Visualization and Integrated Discovery (DAVID, http://david.abcc.ncifcrf.gov/home.jsp) with the cut-off criterion of false discovery rate (FDR) < 0.05. The GO terms were identified in biological process (BP), cellular component (CC) and molecular function (MF) categories. The regulatory relationships for targets genes that simultaneously involved in enriched functions and pathways were selected to construct miRNA-target gene regulatory network, which was visualized using Cytoscape (Version 3.1.1, National Institute of General Medical Sciences, Bethesda, MD, USA, https://cytoscape.org/).

### 4.5. Flow Cytometry

PANC-1 and MIA-PaCa-2 cells were cultured in DMEM supplemented with 10% FBS, which was then dissociated with trypsin solution. After dissociated into single cells, PANC-1 and MIA-PaCa-2 were washed with PBS and counted. Then cells were resuspended in incubation buffer (PBS supplied with 3% FBS) at the concentration of 1 × 10^7^ cells/mL. APC-conjugated anti-human CD44 (311117, BioLegend, San Diego, CA, USA), Alexa Fluor 488-conjugated anti-human EpCAM (324209, BioLegend) and PE-conjugated anti-human CD24 (311105, BioLegend) were added according to the manufacturer’s instructions and incubated at 4 °C protected from light. After 30 min incubation, the cells were washed twice and analyzed on a flow cytometer (FACS Aria II, Becton Dickinson, San Jose, CA, USA).

### 4.6. Immunocytochemistry (ICC)

PANC-1 and MIA-PaCa-2 cells were cultured in DMEM supplemented with 10% FBS, which was then dissociated with trypsin solution. After dissociated into single cells, PANC-1 and MIA-PaCa-2 were washed with PBS and counted. Then cells were seeded into 96-well plate (655090, Greiner, Ludwigsburg, Germany) at the concentration of 8000 cells/well. After cultured overnight, the cells were fixed with 4% paraformaldehyde at room temperature for 10 min. Then the cells were blocked with incubation buffer (PBS supplied with 3% FBS) for 1 h at room temperature. Following that, the cells were also stained with above antibodies at the concentration of 1 μg/mL and incubated at 4 °C protected from light. After 1 h incubation, the cells were washed twice and stained with 5 μg/mL Hoechst 33,342 for 10 min at room temperature. Then the plate was imaged and analyzed with an Operetta CLS high-content imaging system (PerkinElmer Inc., Fremont, CA, USA).

### 4.7. Sphere Formation Culture

Originally, PANC-1 and MIA-Paca-2 cells were cultured in DMEM supplemented with 10% FBS, then dissociated with trypsin solution. Dissociated single cells were transported to Falcon 5 mL polystyrene test tubes and washed twice with phosphate buffer solution (PBS). The cells were then analyzed on a flow cytometer (FACS Aria II, Becton Dickinson, San Diego, CA, USA). Then, PANC-1 and MIA-Paca-2 cells were separately sorted into ultra-low cluster 96-well plate with 200 μL sphere formation medium (SFM) added per well at the concentration of 100 cells/well. The SFM consisted of DMEM/F12 medium (11330-032, Gibco) supplied with 20 ng/mL epidermal growth factor (EGF, 236-EG-200, R&D Systems, Minneapolis, MN, USA), 20 ng/mL basic fibroblast growth factor (bFGF, 233-FB-025, R&D Systems), B27 supplement (17504044, Gibco) and N2 supplement (17502048, Gibco). Subsequently, cells were cultured at 37 °C in a 5% CO_2_ humidified environment and imaged with an Operetta CLS high-content imaging system (PerkinElmer) every other day.

### 4.8. miRNA Mimics Transfection

miRNA mimics and miRNA mimic negative controls were purchased from Guangzhou Ribo-Bio (Guangzhou, China). PANC-1 cells (3 × 10^5^) were dissociated into single cells, seeded onto six-well plates, and cultured overnight. Cells were then transfected with miRNA mimics (miR-7-5p mimic, miR-135b-5p mimic, let-7d mimic, and negative control mimic) at the concentration of 50nM and the mixture of three mimics (50 nM miR-7-5p mimic, 50 nM miR-135b-5p mimic, and 50 nM let-7d mimic) using Lipofectamine 2000 according to the manufacturer’s instruction (Invitrogen, Thermo Fisher Scientific, Carlsbad, CA, USA). Firstly, dilute miRNA mimic in 50 μL Opti-MEM I Reduced Serum Medium and mix gently. Secondly, dilute 1 μL lipofectamine 2000 in 50 μL Opti- MEM I Reduced Serum Medium, mix gently and incubate for 5 min at room temperature. Thirdly, combine the diluted miRNA mimic with the diluted lipofectamine 2000, mix gently and incubate for 20 min at room temperature. Fourthly, add the mimic-lipofectamine 2000 complexes to each well containing cells and medium and mix gently by rocking the plate back and forth. Then, incubate the transfected PANC-1 cells in 96-well plate and cultured at 37 °C in 5% CO_2_ humidified environment for further analysis.

## Figures and Tables

**Figure 1 ijms-20-04473-f001:**
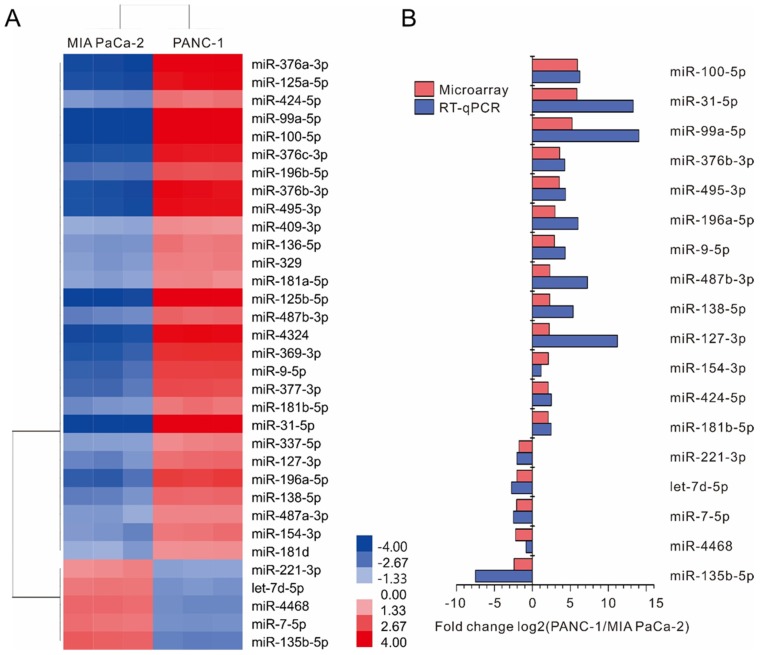
miRNA expression profile in MIA PaCa-2 and PANC-1 cell lines. (**A**) Hierarchical clustering result of 33 differentially expressed miRNAs. Each column corresponds to a single microarray, whereas each row indicates the expression profile of a single gene. The cluster analysis shows a clear separation between the two groups of samples. Colored bars indicate the range of fold change. (**B**) Validation of miRNA microarray data was done for 18 differentially expressed miRNAs using real-time RT-PCR. Data represent the log2-transformed fold change of expression between two pancreas ductal adenocarcinoma cell lines. U6 was used as an internal miRNA control. The qPCR values were displayed in blue and the array results in red.

**Figure 2 ijms-20-04473-f002:**
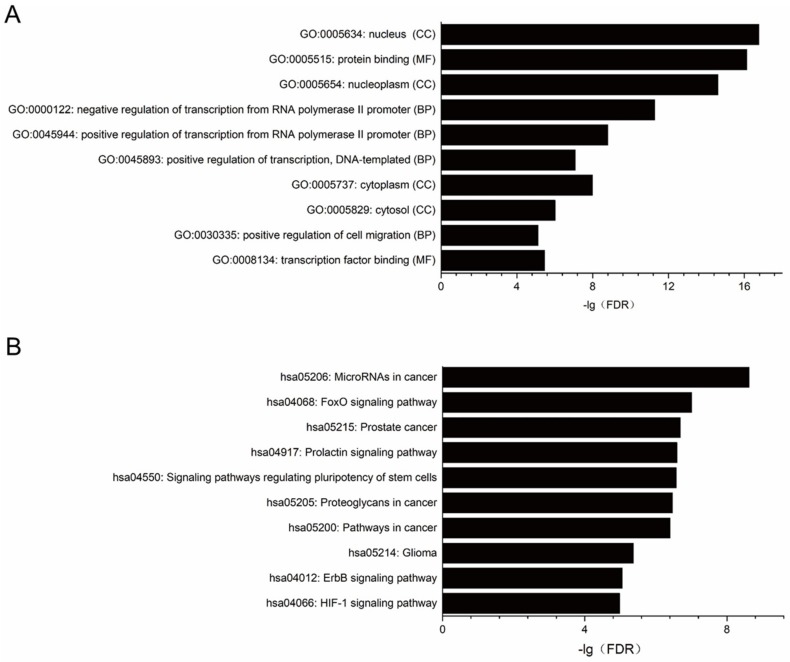
Enrichment analysis of GO function and pathways for predicted miRNAs targets significantly different expressed in PDAC cell lines. (**A**) GO function enrichment analysis. (**B**) KEGG pathway enrichment analysis. Only the top ten GO terms and pathways were shown in this figure. FDR < 0.05 was set as criteria for analysis.

**Figure 3 ijms-20-04473-f003:**
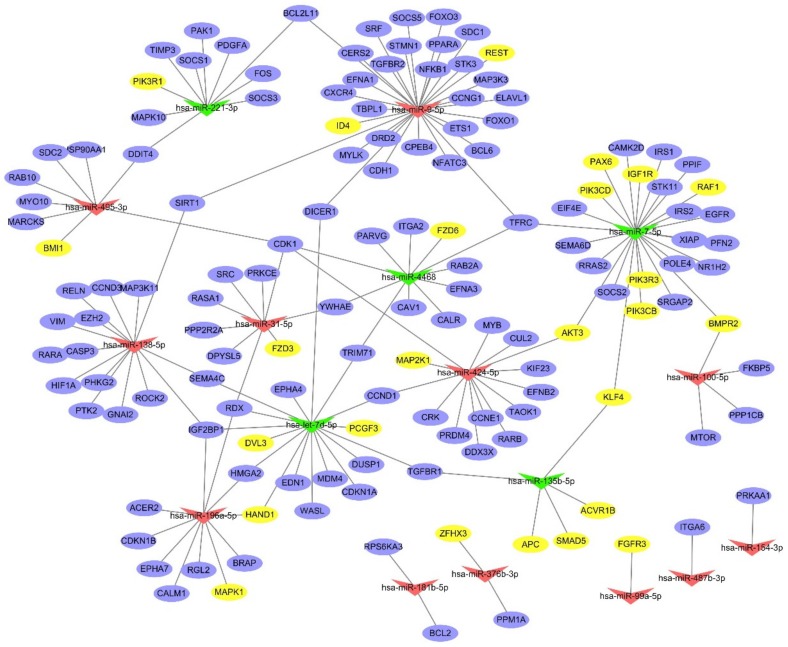
The regulatory network for differentially expressed miRNAs. The purple nodes represent the target genes. The yellow nodes represent those were enriched in the signaling pathways regulating the pluripotency of stem cells. The red triangles indicate the up-regulated miRNAs, while the green triangles indicate the down-regulated miRNAs. The lines show the potential regulatory relationships between the miRNAs and target genes.

**Figure 4 ijms-20-04473-f004:**
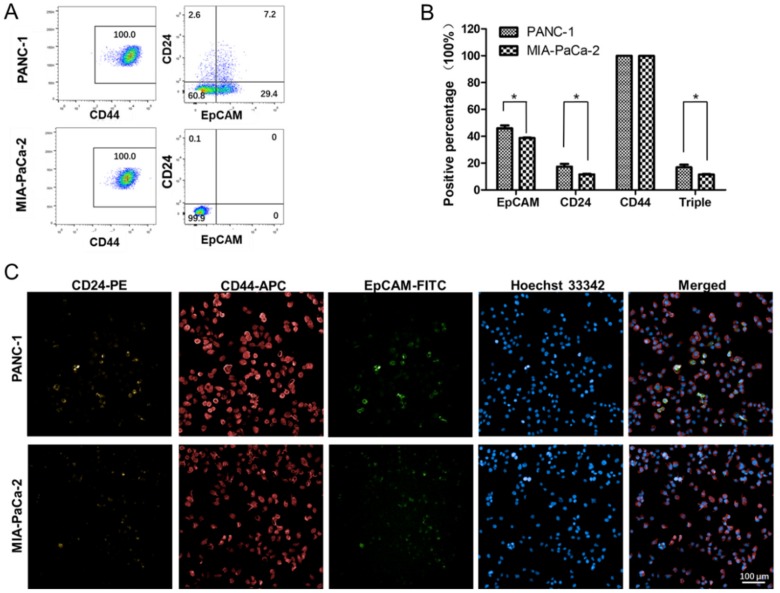
Analysis of pancreatic cancer stem cell surface markers CD24, CD44, and EpCAM in PANC-1 and MIA-PaCa-2. The presence of CD24, CD44, and EpCAM were determined by flow cytometry and immunocytochemistry examination. (**A**) The percentage of triple-positive cells was 7.2% in PANC-1 and 0% in MIA-PaCa-2 through flow cytometry analysis. (**B**) In PANC-1 analyzed with immunocytochemistry examination, the percentage of CD24^+^, EpCAM^+^, and CD24^+^CD44^+^EpCAM^+^ cells was 17.3 ± 3.7%, 46.0 ± 3.5% and 16.8 ± 3.6%, while in MIA-PaCa-2, the percentage was 11.5 ± 0.7%, 38.7 ± 0.5% and 11.4 ± 0.7%, respectively. (**C**) Images of PANC-1 and MIA-PaCa-2 cells stained with anti-human CD44, anti-human EpCAM, and anti-human CD24 antibody. The right panel was the merged images. Data are presented as the mean ± standard deviation (SD) of three independent experiments. * *P* < 0.05 in comparison with the corresponding cells using Student’s *t*-test.

**Figure 5 ijms-20-04473-f005:**
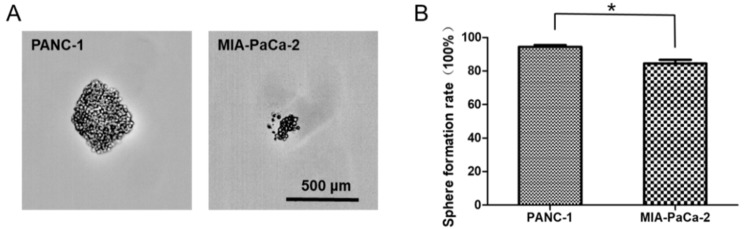
The sphere formation ability of PANC-1 and MIA-PaCa-2. (**A**) Sphere images of PANC-1 and MIA-PaCa-2 cells after nine days culture. (**B**) The difference of sphere formation rate between PANC-1 and MIA-PaCa-2 cells. Data are presented as the mean ± standard deviation (SD) of three independent experiments. **P* < 0.05 in comparison with the corresponding cells using Student’s *t*-test.

**Figure 6 ijms-20-04473-f006:**
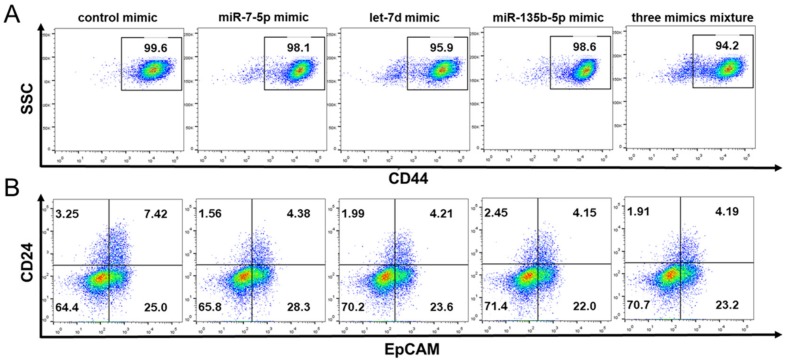
Flow cytometry analysis of pancreatic cancer stem cell surface markers CD24, CD44 and EpCAM in PANC-1 cells transfected with miRNA mimics. (**A**) The percentage of CD44^+^ cells in PANC-1 was 99.6%, transfected with miR-7-5p, let-7d, miR-135b-5p and three mixture mimics was 98.1%, 95.9%, 98.6% and 94.2%, respectively. (**B**) The percentage of CD24^+^EpCAM^+^ cells in PANC-1 was 7.42%, transfected with miR-7-5p, let-7d, miR-135b-5p and three mixture mimics was 4.38%, 4.21%, 4.15% and 4.19%, respectively.

**Figure 7 ijms-20-04473-f007:**
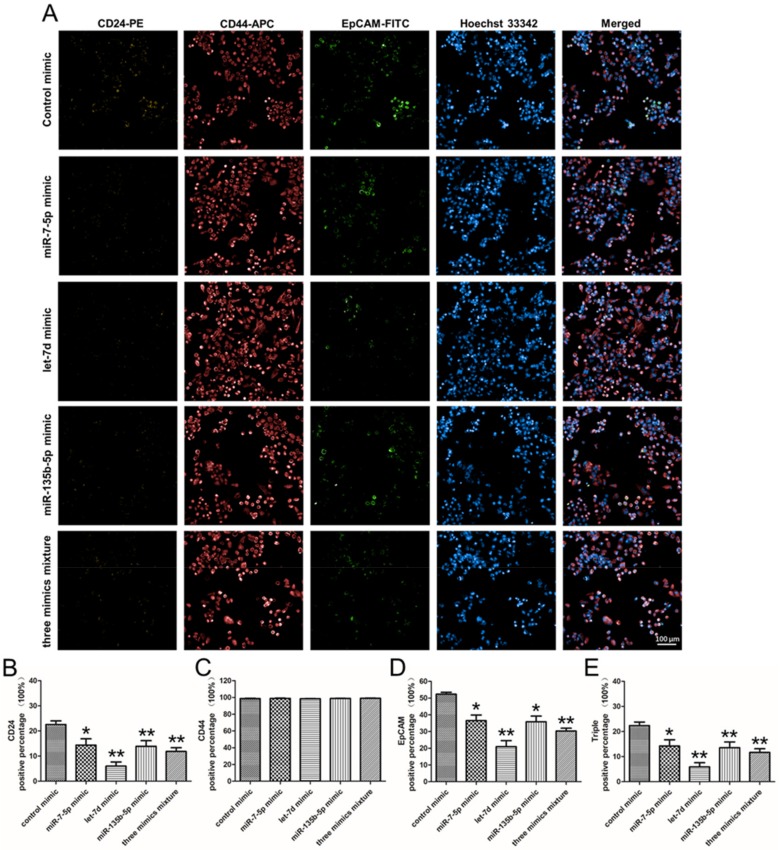
Immunocytochemistry analysis of pancreatic cancer stem cell surface markers CD24, CD44 and EpCAM in PANC-1 cells transfected with miRNA mimics. (**A**) Images of miRNA mimic transfected PANC-1 cells stained with anti-human CD44, anti-human EpCAM and anti-human CD24 antibody. (**B**) The percentage of CD24^+^ cells was 22.6±2.5% (transfected with control mimic), 14.4 ± 4.3% (miR-7-5p mimic), 6.0 ± 2.9% (let-7d mimic), 13.9 ± 3.9% (miR-135b-5p mimic) and 11.9 ± 2.6% (three mimics). (**C**) In the same order, the percentage of CD44^+^ cells was 98.7 ± 0.7%, 98.9 ± 0.6%, 98.5 ± 0.4%, 98.8 ± 0.5% and 99.0 ± 0.6%. (**D**) The percentage of EpCAM^+^ cells was 52.3 ± 1.9%, 36.6 ± 5.7%, 21.0 ± 6.2%, 35.8 ± 5.9% and 30.3 ± 2.8%, follow the same order. (**E**) The percentage of CD24^+^CD44^+^EpCAM^+^ cells was 22.4 ± 2.4%, 14.2 ± 4.3%, 5.9 ± 2.9%, 13.6 ± 3.9% and 11.7 ± 2.5%, in the order above. Data presented as the mean ± standard deviation (SD) of three independent experiments. * *P* < 0.05 in comparison with the corresponding cells using Student’s *t*-test. ** *P* < 0.01 in comparison with the corresponding cells using Student’s *t*-test.

**Figure 8 ijms-20-04473-f008:**
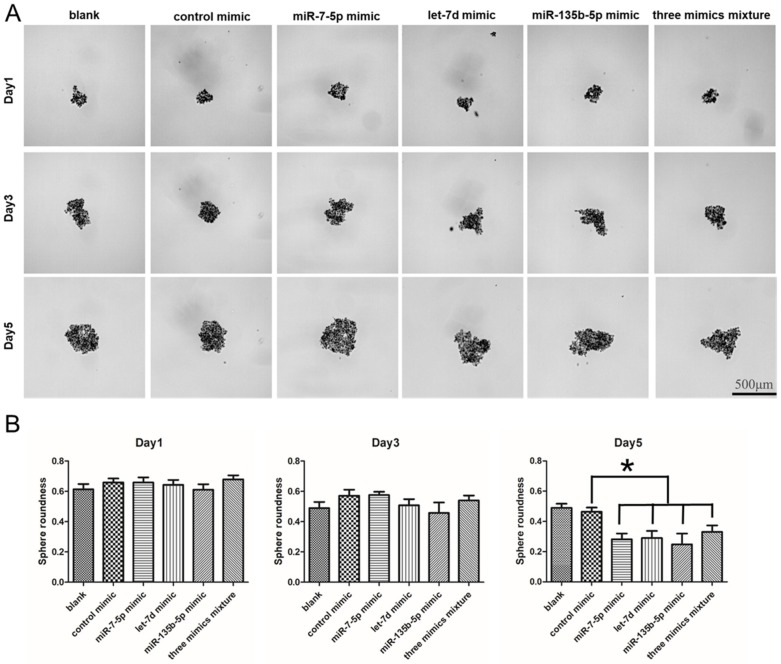
The sphere formation ability of PANC-1 treated with miRNA mimics. (**A**) Sphere images of different treated groups of PANC-1 on day1, 3 and 5. (**B**) The differences of sphere roundness between different treated groups of PANC-1 on day1, 3 and 5. Data are presented as the mean ± standard deviation (SD) of three independent experiments. * *P* < 0.05 in comparison with the corresponding cells using Student’s *t*-test.

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
