# Peer review of "Differentially Expressed microRNAs in MIA PaCa-2 and PANC-1 Pancreas Ductal Adenocarcinoma Cell Lines are Involved in Cancer Stem Cell Regulation"

_ijms, 2019, doi:10.3390/ijms20184473_

Round 1

Reviewer 1 Report

Shen and colleagues presented a research article aimed at establishing the involvement of microRNAs in the regulation of cancer stem cells by analyzing two pancreas ductal adenocarcinoma cell lines, MIA PaCa-2 and PANC-1. In particular, the authors performed some functional experiments and molecular evaluations on the two cell lines in order to establish which de-regulated miRNAs are responsible for the acquisition of surface markers of stemness and spheroid capability.
Overall, the manuscript is well written and the experimental setting appropriate, however, it is not clear the rationale of the study and the choice of the two cell lines as models of CSC. The manuscript needs some minor/major revisions before publication:

1) The paragraph from line 52 to line 59 needs some grammar revisions;
2) The aim of the study is to identify the miRNAs involved in the regulation of CSCs, however, to do this the authors use commercial tumor cell lines (MIA PaCa-2 and PANC-1) which do not represent a good in vitro model of CSCs. Why the authors chose these cell lines? What is the opinion of the authors regarding this critical point?;
3) Why did the authors analyze differentially expressed miRNAs in two tumor cell lines? It would have made more sense to analyze differentially expressed miRNAs in tumor cell lines compared to normal pancreas cells or primary tumor cell and immortalized ones. Please clarify;
4) Several studies demonstrated that the 18 miRNAs up-regulated (13) and down-regulated (5) are involved in the development of several tumors. Please consider to add in the discussion section a paragraph describing other studies where the 18 miRNAs were de-regulated in both solid and hematological tumors. For this purpose see the following papers and other:
- 10.3892/ol.2015.4031
- 10.3390/cancers11050610​
- 10.1371/journal.pone.0217421​
- 10.3892/or.2019.7215​
- 10.18632/oncotarget.5055
5) In the subheading “2.7. The effect of miRNA mimics on the stem cell surface markers expression percentage of PANC-1” of the Results section, please specify if the experiment were performed in triplicate. Furthermore, for the results of Figure 6, please indicate if there are statistical differences among groups;
6) In Discussion section, the authors do not clarify what are the results achieved by their study, and above all what new information this study adds compared to those obtained from other similar studies concerning the molecular characterization of the MIA-PaCa 2 and PANC-1 cell lines and the relationship between miRNAs and CSCs. The Discussion section needs an extensive revision;
7) In the Materials and Methods section, please provide the company and name and specification of the microarray platform used;
8) In the subheading 4.3, 336 did the author evaluated the expression levels of mature miRNAs or stem-loop miRNAs? Please, specify.

Author Response

Comments and Suggestions for Authors

Shen and colleagues presented a research article aimed at establishing the involvement of microRNAs in the regulation of cancer stem cells by analyzing two pancreas ductal adenocarcinoma cell lines, MIA PaCa-2 and PANC-1. In particular, the authors performed some functional experiments and molecular evaluations on the two cell lines in order to establish which de-regulated miRNAs are responsible for the acquisition of surface markers of stemness and spheroid capability. 
Overall, the manuscript is well written and the experimental setting appropriate, however, it is not clear the rationale of the study and the choice of the two cell lines as models of CSC. The manuscript needs some minor/major revisions before publication:

The paragraph from line 52 to line 59 needs some grammar revisions;

We revised this paragraph from line 52 to line 59 as follows:

MIA PaCa-2 and PANC-1 are two commonly used tumor cell lines for in vitro study of PDAC carcinogenesis [13]. They are easily accessible, reliable, and less problematic compared to the primary culture of tumors. According to Deer et al [14], both MIA PaCa-2 and PANC-1 are poorly differentiated, and PANC-1 is derived from a patient with metastasis while the metastatic state of the patient from whom MIA PaCa-2 is derived is unclear. The invasive assays of these two cell lines conducted by different research groups do not produce a consensus and solid result. However, in tumorigenicity assay, MIA PaCa-2 tumors are larger than PANC-1 when Severe Combined Immunodeficiency (SCID) mice receive intraperitoneal injections of equal dose of tumor cells. Moreover, their morphological and genetic characteristics are also well studied [13], suggesting these cell lines are ideal tools for pancreatic cancer research in vitro.

2) The aim of the study is to identify the miRNAs involved in the regulation of CSCs, however, to do this the authors use commercial tumor cell lines (MIA PaCa-2 and PANC-1) which do not represent a good in vitro model of CSCs. Why the authors chose these cell lines? What is the opinion of the authors regarding this critical point?

Excellent questions. Indeed, the commercial tumor cell lines are not ideal to be used as in vitro model for CSCs. Yet we would like to point out that to identify miRNA involvement in CSC regulation is NOT our initial goal of the study; actually, the link between the screened miRNAs and stemness just emerged out of expectation later on. As we mentioned in the paragraph from line 66 to line 67, “In this work, we attempt to seek the heterogeneity between different cancer cell lines, as “precision therapeutics” has specific concern on the individual characteristics.” Therefore, we screened the differentially expressed miRNAs between MIA PaCa-2 and PANC-1 cell lines. And interestingly, we found the target genes of several differentially expressed miRNAs (miR-7-5p, miR-135b-5p and let-7d) were enriched in the signaling pathways regulating the pluripotency of stem cells. Is the result a hint to probable stemness difference of the two PANC cell lines? To seek the answer, we move on to further analyze the CSCs characteristics (surface markers and tumor sphere formation) of these two cell lines.

Of course, a plain reason why we choose these two cell lines is availability. After all, they are easily accessible, reliable, and less problematic compared to the primary culture of tumors. And well fit for the purpose of heterogeneity study. Furthermore, the morphological and genetic characteristics of these two cell lines (MIA PaCa-2 and PANC-1) have been well studied (Ref. 14). For example, MIA PaCa-2 cells revealed the presence of two distinct morphological patterns (large cells and small cells) and expressed immunohistochemical markers of CK5.6, AE1/AE3, E-cadherin, vimentin and chromogranin A but not CD56, while PANC-1 cells revealed the presence of three distinct morphological patterns (large cells, intermediate cells and small cells) and expressed CK5.6, AE1/AE3, vimentin, chromogranin A, CD56 and SSTR2 but not E-cadherin, synaptophysin or NTR1 and synaptophysin. We speculated these results are probably useful when we selected miRNAs for function research at the study design stage. Thus, we chose these cell lines for our research.

3) Why did the authors analyze differentially expressed miRNAs in two tumor cell lines? It would have made more sense to analyze differentially expressed miRNAs in tumor cell lines compared to normal pancreas cells or primary tumor cell and immortalized ones. Please clarify;

Previous study (Ref. 15) have compared the dysregulated miRNAs between 2 non-transformed human pancreatic ductal epithelial cell lines (HPDE and HPNE) as “controls” and 21 pancreatic cancer cell lines (AsPc1, BxPc3, CAPAN1, CAPAN2, COLO357, E3LZ10.7, L3.6PL, MiaPaCa2, Panc-1, Panc-2.03, Panc-327, PK8, PK9, PL3, PL5, PL11, Su86.86, XPA1, XPA2, XPA3, and XPA4), and have demonstrated miR-200 family and the miR-17-92 cluster are uniformly overexpressed in PDAC cell lines to enhance cell proliferation. Although only 321 human miRNAs are monitored in the custom microarray, this is still the most comprehensive article of miRNA screening for so many PDAC cell lines, including normal pancreas cells, primary tumor cell and immortalized ones. In our study, we focused on two well studied PDAC cell lines (MIA-PaCa 2 and PANC-1) (Ref. 14) to seek the heterogeneity between them, as “precision therapeutics” has specific concern on the individual characteristics. Interestingly, we have demonstrated the PANC-1 cell line possesses higher “stemness” feature, and overexpression of these three miRNAs (miR-7-5p, let-7d and miR-135b-5p) result in CSCs characteristics inhibition. We hope our preliminary results may lead to a strategy to treat PDAC patients with more stem-like cancer cells.

4) Several studies demonstrated that the 18 miRNAs up-regulated (13) and down-regulated (5) are involved in the development of several tumors. Please consider to add in the discussion section a paragraph describing other studies where the 18 miRNAs were de-regulated in both solid and hematological tumors. For this purpose see the following papers and other:
- 10.3892/ol.2015.4031
- 10.3390/cancers11050610​
- 10.1371/journal.pone.0217421​
- 10.3892/or.2019.7215​
- 10.18632/oncotarget.5055

We have added a paragraph describing 18 dysregulated miRNAs in the discussion section as follows:

In this study, we have confirmed 18 (13 up- and 5 down-regulated) differentially expressed miRNAs between MIA PaCa-2 and PANC-1 cell lines, and have discovered these miRNAs are also involved in the development of several tumors. For example, miR-7-5p has been found down-regulated in glioblastoma multiforme (GBM) [20], while up-regulated in oral cancer [21]. Meanwhile, miR-99a-5p and miR-196a-5p are strongly associated with the presence of the tumor via the analysis of the TCGA HNSC dataset [21]. In comparison of cancerous and noncancerous tissues of PDACs, miR-31-5p and miR-100-5p show up-regulation and their functions are to promote proliferation, invasion and migration, or inhibit proliferation and increase sensitivities to cisplatin [6, 22]. In addition, miR-181b-5p is up-regulated in colorectal cancer and breast cancer by targeting programmed cell death 4 (PDCD4) and Bim, respectively [23、24]. Furthermore, its expression is also probably correlated with burkitt's lymphoma (BL) [25].  

5) In the subheading “2.7. The effect of miRNA mimics on the stem cell surface markers expression percentage of PANC-1” of the Results section, please specify if the experiment were performed in triplicate. Furthermore, for the results of Figure 6, please indicate if there are statistical differences among groups;

We did not perform this experiment in triplicate. Here the FACS results are qualitative in nature and requires validation by other means. In this paper, we confirmed it by immunocytochemistry, which is quantitative and performed in triplicate. In many research papers (10.1158/1078-0432.ccr-19-0545, 10.1038/cmi.2016.38, 10.1080/19420862.2019.1618673, 10.1158/1078-0432.ccr-17-1952, 10.1080/19420862.2019.1574520), the FACS experiments did not perform in triplicate and provide qualitative result.

6) In Discussion section, the authors do not clarify what are the results achieved by their study, and above all what new information this study adds compared to those obtained from other similar studies concerning the molecular characterization of the MIA-PaCa 2 and PANC-1 cell lines and the relationship between miRNAs and CSCs. The Discussion section needs an extensive revision;

Thanks for the advice. We added three paragraphs and changed the last paragraphs in the Discussion section to discuss above questions.

The existence of CSCs has been widely accepted to be responsible for tumor aggressiveness. In PDAC, CSCs may display additional properties, such as relative drug resistance and enhanced invasion and migration. Emerging evidences suggest that miRNAs may play a key role in the biology of CSCs. A series of miRNAs has been found aberrantly expressed in PDAC CSCs, including miR-99a-5p, miR-100-5p, miR-125b, miR-192, and miR-429 [29]. In the presented study, we have also detected miR-99a-5p and miR-100-5p with differential expression between MIA-PaCa 2 and PANC-1 cell lines. Many previous studies have demonstrated that various miRNAs participate in promoting or inhibiting stemness. For instance, the miR-200 family members repress the expression of stem cell factors, Y–box 2 (Sox2) and Kruppel-like factor 4 (Klf4). The EMT-activator zinc finger E-box binding homeobox 1 (ZEB1) represses the expression of stemness-inhibiting miR-203. These results show that the ZEB1-miR-200 axis probably is a novel therapeutic target against PDAC CSCs [11, 30]. Several other studies also have identified some miRNAs with important roles in cancer stemness in PDAC, such as miR-106a, miR-30a, miR-744 and miR-1246 [31-34]. Moreover, let-7c is shown to decrease pancreatic cancer initiating cell growth by posttranscriptional activation of Numbl and indirect inhibition of Notch [35], which is similar to our results that let-7d mimic partially reduces CSC surface markers and inhibits tumor sphere formation. In other cancers, Sanchez-Diaz et al. have verified the ability of tumor sphere formation is impaired when the expression of hsa-miR-135b-5p is suppressed in SK-N-BE(2) (neuroblastoma) cells [36], the same as we observed in PDAC PANC-1 cells.

Previously, microscopic observation of PANC-1 and MIA PaCa-2 cells have revealed there are two distinct morphological patterns (large cells and small cells) in MIA PaCa-2, while three distinct morphological patterns (large cells, intermediate cells and small cells) in PANC-1 [13]. Furthermore, it is confirmed via immunohistochemistry analysis that E-cadherin expressed in MIA PaCa-2 but not in PANC-1. Since increased expression of E-cadherin is associated with improved survival in several tumor types [46], the behavior of PANC-1 is considered more aggressive with a greater metastasizing potential. These results indicate that PANC-1 is more heterogenous and invasive than MIA PaCa-2, which is in agreement to our results that PANC-1 cells own more cancer stemness.

miRNAs have been viewed as possible therapeutic targets since these molecules affect multiple gene expression and cellular pathways. For instance, transfection of miR-200c and miR-200a mimics into pancreatic CSCs could reduce their colony formation, invasiveness, and resistance to chemotherapy by regulating EMT [47]. When miRNA-205 and miR-7 were transfected into pancreatic cells resistant to gemcitabine, the expression of tubulin beta 3 class III (TUBB3) and p21 activated kinase 1 (Pak-1) were reduced, respectively, and the CSC population was also decreased [48]. These researches support our view that targeting miRNAs could become a novel strategy for killing CSCs, which may become a powerful assistance to the treatment of patients diagnosed with PDAC.

To find miRNA dysregulation in cancers through comparing cancerous vs. normal cells, tissues, or plasma is a conventional way to analyze the tumor-related signal pathways. Nonetheless, inter-cell/patient variability of miRNA expression and signaling pathway is rarely investigated. In this work, we have demonstrated that in pancreatic ductal adenocarcinoma cell lines, PANC-1 cells exhibit higher stemness feature than MIA-PaCa-2 cells, which is probably the consequence of differential regulation of specific miRNAs. Confirmatory assays using the corresponding miRNA mimics show inhibition of the CSC-related characteristics in PANC-1. These findings may offer new therapeutic targets for precise and personalized treatment of patients, or offer an alternative approach to enhance the efficacy of current therapies. For instance, to suppress tumor growth by restoring miRNA expression to kill CSCs. However, the study of the above miRNA-based therapies is still in its infancy, more researches are required to verify the effectiveness of specific therapeutic targets for individual differences.

7) In the Materials and Methods section, please provide the company and name and specification of the microarray platform used;

Thanks for reminding us. We added the microarray information in the Materials and Methods section as follows:

The miRNA microarray platform was used as previously described using a fluorescent label-free strategy with good data reproducibility, high specificity and sensitivity [16]. These microarrays were custom-made by Phalanx Biotech Group (Hsinchu, Taiwan, China) and each probe printed in quadruplicate. The method of chip activation was followed the company's protocol, and the procedure of chip hybridization was described as follows.

8) In the subheading 4.3, 336 did the author evaluated the expression levels of mature miRNAs or stem-loop miRNAs? Please, specify.

We evaluated the expression levels of mature miRNAs and changed the sentences in the subheading 4.3, 336 as follows:

Real-time qPCR was performed to evaluate the expression levels of mature miRNAs on a Roche LightCycler System (Roche Diagnostics, Rotkreuz, Switzerland) using SYBR Premix Ex TaqTM II (TaKaRa, RR820A).

Reviewer 2 Report

This paper discovers differently expressed miRNAs in MIA PaCa-2 2 and PANC-1 pancreas ductal adenocarcinoma (PDAC) cell lines. PDAC is a deadly malignant disease. MIA PaCa-2 and PANC-1 cell lines are primary tumors used to study PDAC carcinogenesis, and thus this paper focuses on exploring differently expressed miRNAs between these two cell lines.

Comments.

line 56. ‘MIA PaCa-2 appeared to result in tumor with larger than PANC-1’. This is not clear. lines 61 and 62. You said that only 321 human 61 miRNAs are monitored in the custom microarray [15]. But the published year of this reference is 2009. Is there any other study discussing the topic after that? line 84. Are these 33 miRNAs differently expressed between PDAC and normal controls? line 100. Only 18 miRNAs were done for PCR. How about the other 15 miRNAs? lines 134 and 135. Explain this sentence ‘indicating that these stem cell related genes are probably upregulated 134 in PANC-1 cell lines’. line 16. ‘thus’ → ‘and thus’

Author Response

This paper discovers differently expressed miRNAs in MIA PaCa-2 2 and PANC-1 pancreas ductal adenocarcinoma (PDAC) cell lines. PDAC is a deadly malignant disease. MIA PaCa-2 and PANC-1 cell lines are primary tumors used to study PDAC carcinogenesis, and thus this paper focuses on exploring differently expressed miRNAs between these two cell lines.

Comments.

1, line 56. ‘MIA PaCa-2 appeared to result in tumor with larger than PANC-1’. This is not clear.

Thanks for the suggestion. As this sentence explained unclearly, we changed it as follow:

However, in tumorigenicity assay, MIA PaCa-2 tumors are larger than PANC-1 when Severe Combined Immunodeficiency (SCID) mice receive intraperitoneal injections of equal dose of tumor cells [13]

2, lines 61 and 62. You said that only 321 human 61 miRNAs are monitored in the custom microarray [15]. But the published year of this reference is 2009. Is there any other study discussing the topic after that?

This published paper (Ref. 15) is most relevant with PDAC cell lines and miRNAs, although the published year is 2009. In this reference, 2 non-transformed human pancreatic ductal epithelial cell lines (HPDE and HPNE) as “controls” and 21 pancreatic cancer cell lines (AsPc1, BxPc3, CAPAN1, CAPAN2, COLO357, E3LZ10.7, L3.6PL, MiaPaCa2, Panc-1, Panc-2.03, Panc-327, PK8, PK9, PL3, PL5, PL11, Su86.86, XPA1, XPA2, XPA3, and XPA4) were screened by human miRNA microarray with only 321 human miRNAs. However, this is still the most comprehensive article of miRNA screening for so many PDAC cell lines. Today we search the literature database and find some other topics, such as certain miRNAs have been identified as potential prognostic biomarkers in PDAC patients (10.1186/s40364-018-0131-1, 10.1016/j.ajpath.2018.10.005, 10.1053/j.gastro.2013.10.010) and miRNAs and their molecular targets may serve as targets to enhance sensitivity to chemotherapy and reduce metastatic spread (10.1371/journal.pone.0106343). As far as we know, there is no new article discussed about miRNA screening in PDAC cell lines.

3, line 84. Are these 33 miRNAs differently expressed between PDAC and normal controls?

In the subheading 2.1, these 33 miRNAs differentially expressed between MIA PaCa-2 and PANC-1 cell lines. In this work, we screened the differentially expressed miRNAs between MIA PaCa-2 and PANC-1 cell lines (two cancer cell lines) and attempted to seek the heterogeneity between different cancer cell lines, as “precision therapeutics” has specific concern on the individual characteristics.

4, line 100. Only 18 miRNAs were done for PCR. How about the other 15 miRNAs?

Thirty-three differentially expressed miRNAs identified from microarray were further selected for validation by qPCR, and the expression patterns of only 18 miRNAs were consistent with the microarray results. That means the expression patterns between microarray and qPCR were different for the other 15 miRNAs. Since microarray is a high-throughput, yet less precise platform, it is better to validate these results by a low-throughput, high accuracy assay (qPCR).

Furthermore, qPCR is considered as a “gold standard” in the quantitation of gene expression, it is an ideal technique for verification of miRNA microarray results.

5, lines 134 and 135. Explain this sentence ‘indicating that these stem cell related genes are probably upregulated 134 in PANC-1 cell lines’.

miRNAs act as negative regulators in the translational control of gene expression. By base-pairing, miRNAs interact with the 3′ UTR of the target mRNAs to facilitate the recruitment of a ribonucleoprotein complex that either blocks cap-dependent translation or triggers target mRNA deadenylation and degradation. For the reason described above, we speculated the target genes probably upregulated as their miRNAs were downregulated. Considering this sentence explained unclearly, we changed it to this “As these three miRNAs were down-regulated in PANC-1 cell lines, the expression level of these stem cell related genes were probably elevated in PANC-1 cell lines since miRNAs were known as negative regulator of their target genes.”

6, line 16. ‘thus’ → ‘and thus’ 

Thanks for the advice, we change it.

Round 2

Reviewer 2 Report

This version has addressed all my comments.